# Hemagglutinin Gene Variation Rate of H9N2 Avian Influenza Virus by Vaccine Intervention in China

**DOI:** 10.3390/v14051043

**Published:** 2022-05-13

**Authors:** Ying Cao, Haizhou Liu, Di Liu, Wenjun Liu, Tingrong Luo, Jing Li

**Affiliations:** 1State Key Laboratory for Conservation and Utilization of Subtropical Agro-Bioresources & Laboratory of Animal Infectious Diseases, College of Animal Sciences and Veterinary Medicine, Guangxi University, Nanning 530004, China; caoyingor@163.com; 2Computational Virology Group, Center for Bacteria and Viruses Resources and Bioinformation, Wuhan Institute of Virology, Chinese Academy of Sciences, Wuhan 430071, China; liuhz@wh.iov.cn (H.L.); liud@wh.iov.cn (D.L.); 3CAS Key Laboratory of Pathogenic Microbiology and Immunology, Institute of Microbiology, Chinese Academy of Sciences, Beijing 100101, China; 4Savaid Medical School, University of Chinese Academy of Sciences, Beijing 100039, China

**Keywords:** avian influenza virus, H9N2 subtype, vaccine intervention, genetic diversity, variation rate

## Abstract

H9N2 subtype avian influenza virus (AIV) is widespread globally, with China being the main epidemic center. Inactivated virus vaccination was adopted as the main prevention method in China. In this study, 22 hemagglutinin (*HA*) sequences were obtained from all inactivated vaccine strains of H9N2 subtype AIVs in China since its introduction. A phylogenetic analysis of the vaccine sequences and *HA* sequences of all published H9N2 subtype AIVs was conducted to investigate the relationship between vaccine use and the virus genetic diversity of the virus. We found that during 2002–2006, when fewer vaccines were used, annual genetic differences between the *HA* sequences were mainly distributed between 0.025 and 0.075 and were mainly caused by point mutations. From 2009 to 2013, more vaccines were used, and the genetic distance between sequences was about 10 times greater than between 2002 and 2006, especially in 2013. In addition to the accumulation of point mutations, insertion mutations may be the main reason for the large genetic differences between sequences from 2009 to 2013. These findings suggest that the use of inactivated vaccines affected point mutations in the *HA* sequences and that the contribution of high-frequency replacement vaccine strains to the rate of virus evolution is greater than that of low-frequency replacement vaccine strains. The selection pressure of the vaccine antibody plays a certain role in regulating the variation of *HA* sequences in H9N2 subtype AIV.

## 1. Introduction

In 1966, the first H9N2 subtype avian influenza virus (AIV) was isolated from turkeys in the United States [1], and since then, the H9N2 subtype AIVs has been reported in many countries and regions. In the mid-1970s, the H9N2 subtype AIV was first reported in Hong Kong [2], followed by the first outbreak in Guangdong Province of China in 1994 [3]. Since then, the H9N2 subtype AIV has become the main epidemic virus in poultry, such as chickens and ducks, and has spread to Europe, Africa, Asia, and other places [3,4]. The transmission of H9N2 subtype AIV can be seen in almost every country and region in the world.

H9N2 subtype AIV is a low pathogenic AIV, but it can delay the growth of poultry, reduce egg production, and lead to poor eggshell quality [5]. Furthermore, the infection of H9N2 subtype AIV in poultry is also related to coinfection or secondary infection with the infectious bronchitis virus, *Mycoplasma pneumoniae*, and other secondary respiratory pathogens, which can lead to a high mortality rate and cause huge economic losses to the poultry industry in China [6,7]. It has been suggested that the H9N2 subtype AIV can provide internal segments for other subtypes of influenza viruses, and those infected with H9N2 subtype AIV are the “incubators” for influenza virus infections in humans [8]. For instance, the H9N2 subtype AIV was the source of internal segments of the H7N9 subtype AIV in the 2013 outbreaks in Shanghai, Beijing, and other provinces [9]. This was the first time the virus infected humans and caused deaths.

As an indispensable part of poultry farming in the world, the poultry farming in China has become an important part of modern agriculture and has a profound impact on the lives of residents. At present, the H9N2 subtype AIV is prevalent in poultries worldwide, and China is considered to be the epidemic center [10]. Bi et al. showed that H9N2 has replaced H5N6 and H7N9 viruses as the main subtype of AIVs in chickens and ducks in China [11]. To reduce the impact of virus infection on poultry, commercial inactivated vaccines against the H9N2 subtype of influenza viruses, such as A/chicken/Shanghai/F/1998 (F/98), have been approved in China since 1998. It is undeniable that the use of vaccines has prevented outbreaks and virus transmission, but studies have shown that sporadic infections have occurred in vaccinated chicken farms, and the replication capacity of the new virus has been greatly enhanced [12]. Therefore, it is significant to systematically study vaccine use and the genome diversity of the H9N2 subtype influenza viruses.

In this study, most of the inactivated vaccine strains in China since the introduction of the vaccine were collected, and 22 effective sequences were obtained. The vaccine sequences were combined with the published hemagglutinin (*HA*) sequences of H9N2 subtype AIV, and the genetic distances between sequences in annual data sets were calculated at two representative time periods to explore the effects of vaccine use and viral genetic distances.

## 2. Materials and Methods

### 2.1. Collection and Sanger Sequencing of H9N2 Vaccines

The inactivated vaccine sequence was a gift from the vaccine company (Appendix A). The total RNA of the inactivated vaccine was isolated using the Invitrogen TRIzol reagent (Thermo Fisher Scientific, Waltham, MA, USA) as per the manufacturer’s instructions. Reverse Transcription Polymerase Chain Reaction (RT-PCR) was performed using the H9N2 RNA samples. PCR primers used were as follows: forward primer (Uni-12), 5′-GCTCCACACAGAGCATAAT-3′ and reverse primer (Uni-13), 5′-CTTCTGTTGCCACACTCGTT-3′. The PCR consisted of 35 cycles of a denaturing step at 95 °C for 30 s, an annealing step at 57.5 °C for 30 s, and an extension step at 72 °C for 45 s. PCR products were sent to Sangong Bioengineering Co., Ltd. (Shanghai, China) for sequence determination, and the *HA* sequence of the fire-extinguishing vaccine was obtained by sequence splicing using DNASTAR Lasergene software (version 11.0, DNASTAR, Inc., Madison, WI, USA) [13].

### 2.2. Sequence Collection and Alignment

Sequences of H9N2 subtype AIVs with complete genomes isolated in China from 1979 to 2021 were downloaded from the Global Initiative on Sharing Avian Influenza Data (www.gisaid.org, accessed on 21 November 2021) and the Influenza Virus Resource at the National Center for Biotechnology Information (www.ncbi.nlm.nih.gov, accessed on 21 November 2021). It is important to note that the strains with complete genomes were included in the study to ensure the reliability of the analysis. We believe that strains containing full genomes are theoretically more likely to be strains that have been well-studied and whose sequence quality is more assured. All the sequences downloaded from different databases were filtered out by strains for duplicate sequences and then combined with vaccine sequences. The filtered nucleotide sequences were aligned using the MUSCLE software (version 3.8.1551, Sonoma, CA, USA) [14], manually adjusted to correct slippage errors, and subsequently translated. We then counted all sequences for host (environmental, avian, and human) and geographic origin (country and provinces of mainland China). The nucleotide sequence was then translated into the corresponding protein sequence, and multisequence alignment was performed with the F/98 vaccine strain. The translated protein sequences were divided into different data sets according to the collection time, and representative sequences were then selected for display.

### 2.3. Phylogenetic Analysis

The aligned sequence after editing was analyzed by the IQ-Tree software (version 1.6.12, University of Vienna, Medical University of Vienna, Vienna, Austria) [15] using a maximum likelihood method, and the optimal replacement model was automatically calculated before the evolutionary tree was constructed [16]. The bootstrap value used to test branching reliability was set to 1000, and the optimal evolution model selected was GTR + F + R8.

### 2.4. Structural Prediction of HA Protein

Taking the *HA* sequence of the earliest vaccine strain, F/98, as the target sequence, the three-dimensional structure model of *HA* of this strain was predicted using comparative modeling on the SWISS-MODEL platform (www.swissmodel.expasy.org, accessed on 5 December 2021) [17]. According to the target sequence, the *HA* sequence with the highest global model quality estimation was selected as the template for homology modeling based on HHblits (Ludwig-Maximilians Universität München, Munich, Germany). Its score is expressed as a number between 0 and 1, reflecting the expected accuracy and target coverage of a model constructed using alignment and templates, and the higher the number, the higher the reliability [18].

### 2.5. Calculation of Genetic Distance between Sequences

All sequences were divided into different data sets based on collection time and aligned with the earliest vaccine strains using the MUSCLE software (version 3.8.1551) [14], and only specific locations of the alignment sequences were retained. The R package (version 5.5) of Analysis of Phylogenetics and Evolution (APE) was then used to calculate the genetic distance between sequences, and the model was selected as Jukes–Cantor [19]. The resulting intersequence genetic distances were stored in matrix format.

### 2.6. Statistics of Amino Acid Types at Specific Sites

NovoPro (www.novopro.cn, accessed on 15 December 2021, Shanghai Neupu Biotechnology Co., Ltd., Shanghai, China) was used to translate the intercepted nucleotide sequence data set from different years into amino acid sequences. The translated protein sequences were divided into different data sets according to the collection time, and representative sequences in the comparison were selected for display. Subsequently, the type of amino acids present at each position was generated to create a nucleotide map that could reflect the sequence preference of that position.

## 3. Results

### 3.1. H9N2 Subtype AIV Is Mainly Distributed in East Asia

A total of 28 vaccine strains were obtained (Appendix A), of which two strains had no specific approval time. In this study, 28 inactivated vaccine strains with complete information were sequenced using the Sanger method, and 22 *HA* sequences were obtained. Comparing the vaccine sequences with H9N2 subtype AIV sequences downloaded from the Global Initiative on Sharing Avian Influenza Data and the Influenza Virus Resource at the National Center for Biotechnology Information, it was found that they were mainly distributed in China, followed by Vietnam and Bangladesh (Figure 1A). A total of 1436 sequences from China were downloaded, including 1253 avian, 147 environmental, and 36 human samples. Further statistics on the geographical distribution of H9N2 subtype AIVs in China showed that the proportion of H9N2 subtype AIVs was high in Guangdong, Jiangxi, and Shandong Provinces in eastern China (Figure 1B). It is worth noting that Shandong and Guangdong are both major poultry-producing provinces in China, so it is extremely necessary to pay close attention to the monitoring and timely prevention of AIVs in those areas. Although H9N2 subtype AIV was originally identified in the United States, the epidemic epicenter is now mainly in Asia, with China, Vietnam, and Bangladesh accounting for 80% of the global H9N2 strains (Figure 1C). Among the provinces in mainland China, Guangdong, Jiangxi, Shandong, and Jiangsu together account for 48% of the H9N2 subtype AIVs in China (Figure 1D). The detailed distribution of H9N2 subtype AIVs in different countries and cities is shown in Appendix A. The statistics on the sources of AIVs in different provinces showed that, in some provinces with high distribution of H9N2 subtype AIVs, the virus hosts were mainly avian, with only a small proportion from the environment and humans (Figure 1E). Therefore, the H9N2 subtype AIV in China plays an important role in the global influenza epidemic, and it is necessary to strengthen the research on it.

### 3.2. Phylogenetic Analysis of HA Gene of H9N2 Subtype Influenza Virus in China

We performed a phylogenetic analysis of 1274 *HA* genes in all genome-wide H9N2 AIVs from Chinese poultry, including 22 vaccine sequences with both sequence information and approval time (Figure 2A). The root of the evolutionary tree was selected from the Hong Kong strain A/duck/Hong Kong/784/1979, which was isolated in 1979. In phylogenetic trees, branch length represents base replacement rate and is used as a measure of genomic distance between ancestor and progeny viruses. As can be seen from the tree, the later isolates are generally farther from the root of the evolutionary tree in any particular clade, for example, the distance between the branches representing 2013, 2017, and 2019 and the root node successively increases. The sequences with a closer genetic distance also have closer distribution in the phylogenetic tree. Vaccine sequences are shown in red, as can be seen from the partially enlarged tree on the left, with a longer genetic distance between the vaccine sequence and the virus sequence isolated from the year of its use, such as a longer distance between the 2013 vaccine sequence approved for marketing and the 2013 strain (purple branch). In part, the results reflect the effect of vaccine on virus replication and transmission to some extent. Figure 2B shows the distribution of vaccines approved for use in different years since the implementation of the H9N2 influenza subtype vaccination program in China. The results showed that, from the distribution of vaccines approved for use in different years since the implementation of the H9N2 subtype influenza virus vaccination program (Figure 2B), the number of vaccines used in China was relatively small from 2002 to 2006 (five strains), and the number of vaccines put into use increased from 2009. There is no significant proportional relationship between the number of vaccines approved each year and the number of cases and the genetic distance between viruses. For instance, the number of vaccines in use in 2013 is small, but the *HA* sequences of viruses are not close in evolutionary distance. As we know, it is difficult to assess whether the effects of vaccine selection pressure on strains are visible in the current year or whether there is a cumulative effect. In conclusion, we can speculate that there is some correlation between the evolutionary distance of the H9N2 subtype AIV genome and the continuous use of inactivated vaccines.

### 3.3. Three-Dimensional Structure and Mutation Hotspots of the HA Protein of the F/98 Inactivated Vaccine Strain

The inactivated vaccine, F/98, was the first inactivated vaccine approved for H9N2 subtype influenza virus in China. Figure 3A shows the three-dimensional structure of its predicted HA protein from the front elevation and side elevation. Su et al. [20] inoculated chicken embryos with F/98 inactivated vaccine to study the effect of the selection pressure of the vaccine antibody on the evolution of H9N2 subtype AIVs in chickens. In terms of antigenic mutations, the *HA* gene of the progeny virus, with selection pressure from vaccine antibodies, had mutation at sites 131, 168, 198, 201, and 234, while the *HA* gene of the progeny virus had no selection pressure from vaccine antibodies at sites 114, 224, 198, 234, 281, and 285. It can be observed that the mutation hotspots were mainly distributed in the first 285 amino acids, which were mainly concentrated in the HA1 protein sequence (Figure 3B). These sites are primarily distributed on the outer surface of the protein structure of the three-dimensional structure prediction diagram of the HA protein. To study the evolution rate of the *HA* gene more specifically under vaccine selection pressure and to explore the key loci causing changes in the evolution rate, our subsequent studies on the *HA* gene mainly focused on the 855-nucleotide sequence encoding this amino acid.

### 3.4. Evolutionary Distance between HA Genes from 2009 to 2013 Was about Ten Times That from 2002 to 2006

To understand the relationship between the genetic diversity of AIV and vaccine use, we performed a phylogenetic analysis of the *HA* gene of the H9N2 subtype AIV. Two time periods were selected: the first five years after the inactivated vaccine was introduced to the market from 2002 to 2006, and the second from 2009 to 2013, when the highest number of vaccines was introduced. With the addition of five inactivated vaccine sequences, there are 32 *HA* sequences of H9N2 subtype influenza viruses between 2002 and 2006. In the phylogenetic tree, we selected the *HA* sequence of the F/98 inactivated vaccine as the root. According to the topology of the evolutionary tree, except for the vaccine sequence, the evolutionary distance between other *HA* sequences is within 0.2 (Figure 4A). According to the collection time of the sequence, the distance matrix between the sequences in the data set is calculated with the annual sequence as the data set. The results showed that the genetic distances between the sequences of the data sets before the use of the vaccine did not significantly differ from the years before the vaccine was marketed. In order to clearly illustrate the annual distances between *HA* sequences, we removed the vaccine sequences and presented the distances between the sequences in the form of scatter plots (Figure 4B). It was found that the annual genetic distances between sequences were mainly distributed between 0.025 and 0.075.

A total of ten inactivated vaccine strains were approved to enter the market between 2009 and 2013. Together with genome-wide *HA* sequences downloaded from the database, we performed a phylogenetic analysis on a total of 641 sequences. From the phylogenetic tree (Figure 4C), more sequences of H9N2 subtype influenza viruses were collected from 2009 to 2013 compared to that from 2002 to 2006, and the genetic distance between the sequences was larger (the 2009 vaccine sequence was used as the root node). Correspondingly, the annual distances between *HA* genes calculated after removing the vaccine sequences is shown in Figure 4D. The evolutionary distance between the *HA* genes from 2009 to 2013 was about ten times that from 2002 to 2006, especially in 2013.

### 3.5. HA Gene Mutations Included Both Point and Fragment Mutations from 2009 to 2013

To explore the key loci causing the mutation of *HA* genes under vaccine selection pressure, we translated the nucleotide sequence into the corresponding protein sequence and performed multisequence alignment with the F/98 vaccine strain. The translated protein sequences were divided into different data sets according to the collection time, and representative sequences in the comparison were selected for display. Compared with the results of Hai Long et al., our protein sequence alignment results were consistent with their reported mutational sites related to vaccine selection pressure, such as K131R, A168T, and Q234L [20]. However, the HA protein sequences at sites 114, 201, 281, and 285 were highly conserved for five consecutive years without frequent point mutations, and the dominant amino acids at sites 198 and 224 were inconsistent. In addition to their reported loci, we also found some mutation sites, such as at 66, 71, 87, 92, 93, 107, 166, 182, 183, 213, 244, 253, 282, and 283 (Figure 5A). For the period 2009 to 2013, after the *HA* gene sequences were translated into protein, representative sequences for each year were selected for observation. In addition to the mutation sites, mutations also occurred in the 47, 63, 143, 167, 216, 220, 235, and 254 amino acid sequences compared with those of the F/98 vaccine strain (Figure 5B). Except for the accumulation of point mutations, insertion mutation may be the main reason for the large genetic difference between sequences at this stage. These findings suggest that the use of inactivated vaccines has affected point mutations in the *HA* sequence and that the high-frequency variation of vaccine strains contributed more to the rate of virus evolution than the low-frequency variation of vaccine strains did.

## 4. Discussion

In our study, 22 *HA* sequences represent all the inactivated vaccine strains of H9N2 subtype AIVs in China since its introduction. Phylogenetic analyses of the vaccine sequences and *HA* sequences of all published H9N2 subtype AIVs were conducted to explore the relationship between vaccine use and virus genetic diversity. The influenza genome consists of eight single-stranded, negative-sense RNAs [21]. Among them, the *HA* segment mainly encodes the glycoprotein that the influenza virus recognizes as the host cell surface receptor [22,23], which is the significant surface antigen of influenza A virus and the main target of the host immune response. After infection with AIVs or vaccination, the specific antibody of the HA protein can be produced by the host. During influenza virus interaction with the host, the virus escapes the antibodies by mutating amino acids on the HA segment. Furthermore, the HA1 protein is the main site of antigenic drift due to its large number of epitopes [24]. On the other hand, the influenza virus is a segmented virus, and the study of its internal gene segments is extremely challenging and complex because of the possibility of reassorting events. Moreover, the development of vaccines mainly targets the HA protein of the influenza virus. Therefore, this study focuses on the *HA* gene of H9N2 subtype AIVs.

In 1998, the F/98 strain was isolated in eastern China. Since then, the F/98 strain has spread in eastern China and evolved into multiple branches [25,26]. It is the wild strain corresponding to the first inactivated vaccine approved and put on the market in China. Its mutation sites of continuous passage in chickens have been reported and were used as the reference sequence in our study [20]. Because there is no guarantee that each year’s vaccine represents that year’s dominant virus strain, using sequences from a single year as a basis for determining the relationship between vaccine selection pressure and virus evolution could lead to false positives. In addition, we could not assess whether the effect of vaccine selection pressure on strains was visible in the current year or whether there was an accumulative effect. For instance, the number of vaccines put into use in 2013 was not large, but the evolutionary distance between virus *HA* sequences was big. Therefore, we calculated the genetic distance of the influenza virus *HA* gene for each year and chose a five-year period for our discussion.

The phylogenetic analysis showed that the phylogenetic distance between sequences from 2009 to 2013 was significantly higher than that from 2002 to 2006. The possible reason is that, from 2002 to 2006, a total of three inactivated vaccines entered the market. It is possible that, due to marketing and other reasons, the vaccines did not play a full role, and there was a lag in the immune pressure of the virus. There was a significant increase in the number of inactivated vaccine strains approved for use between 2009 and 2013 with increasing immune pressure on influenza viruses and, thus, more opportunities for the strains to mutate. Similar changes to *HA* gene evolution have also been reported following implementation of the H5N2 low-pathogenic avian influenza vaccine in Mexico [27]. In 1995, a vaccination program was widely implemented for commercial poultry to control the low-pathogenic H5N2 AIVs in Mexico. New lineages emerged after the vaccine was introduced to replace the original viruses. In addition, previous studies have linked the long-term use of the highly pathogenic H5N1 avian influenza vaccine with the emergence of vaccine-resistant field viruses in China, Egypt, Indonesia, and Vietnam [28,29,30]. Antigenic analysis of both the parent and recombinant viruses produced by reverse genetics using hemagglutination inhibition and microneutralization tests showed that the antigenic drift of H5N1 in poultry is driven by multiple mutations occurring primarily at major antigenic sites in the receptor binding subdomain [31].

A total of eight inactivated H9N2 subtype influenza vaccines were put on the market from 2009 to 2010, while only one inactivated vaccine was approved for market entry in 2013. However, the evolutionary distance between sequences in 2013 was about three times the average of the previous four years. Therefore, it is reasonable to support that the introduction of inactivated vaccines might have delayed the effect on the evolution of the virus sequence.

Based on our results, in addition to sites 131, 198, 224, and 234, which have been reported to be associated with immune selection pressure, sites 66–71, 87–93, 107, and 183 were also positive sites for *HA* gene antigen escape mutations of the influenza virus. In particular, the strain isolated in 2013 had a fragment mutation (15 sequences represented by A/chicken/Jiangxi/36291) compared with the F/98 vaccine strain. These results suggest that H9N2 subtype AIVs isolated in China after 2009 were experiencing increasing antigenic drift. It is worth considering whether the possible cause is related to the use of inactivated H9N2 subtype vaccine to some extent. We can see an increase in the *HA* sequence mutation sites as more inactivated vaccines were approved for market entry. Furthermore, a number of insertion mutations were present in virus sequences from 2009 to 2013 compared to the F/98 vaccine strains. Although the phenotypic effects of positive selection and mutation at these antigenic sites are unknown, these sites should be closely monitored to determine the relationship of antigenicity and genetic characteristics. We need to strengthen the monitoring of vaccine use and the virus evolution of the H9N2 subtype AIVs, and these data will help update and use vaccine strains in a more timely and scientific manner.

## Figures and Tables

**Figure 1 viruses-14-01043-f001:**
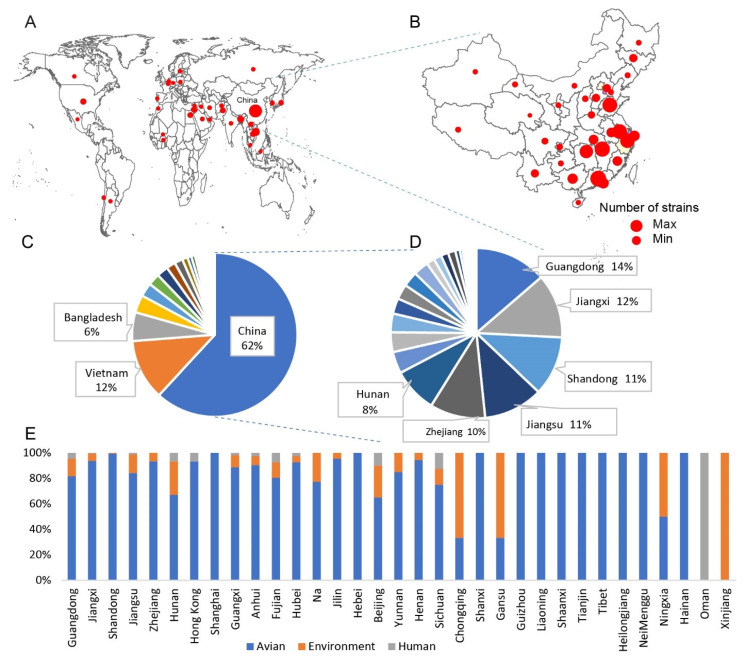
Geographical and host distribution of H9N2 subtype avian influenza virus (AIV). (**A**) Map showing the number of H9N2 cases in the different countries. (**B**) Map showing the number of H9N2 cases in the provinces of China. The diameter of the circle is positively correlated with the number of H9N2 subtype AIVs. (**C**) Proportion of H9N2 subtype AIVs in different countries. Countries with less than 5% influenza viruses are not shown in the chart. (**D**) The proportion of H9N2 subtype AIVs in the different provinces of China. Cities with less than 5% influenza viruses are not shown in the chart. (**E**) Proportion of sources from which H9N2 subtype AIV samples were collected. The main sources are avian, environment, and human.

**Figure 2 viruses-14-01043-f002:**
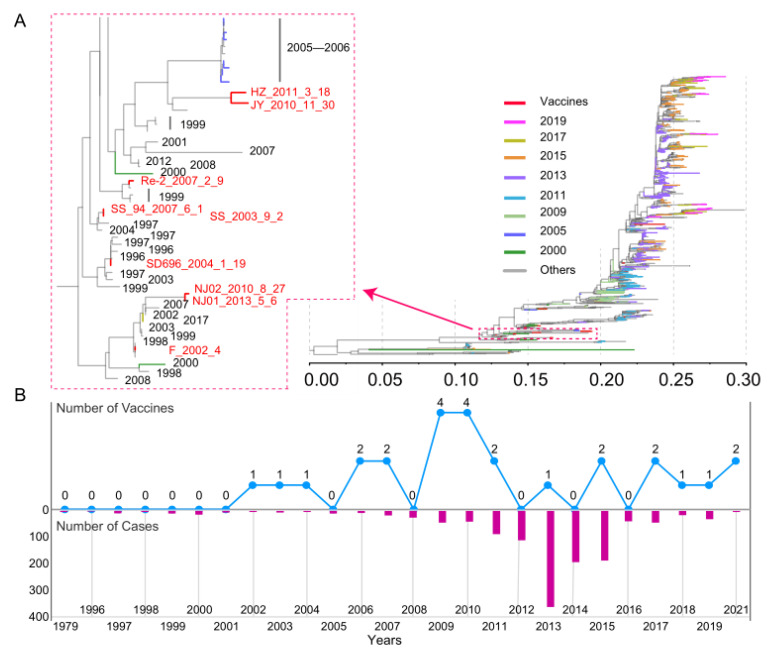
Phylogenetic analysis of the hemagglutinin (*HA*) gene in H9N2 AIVs. (**A**) Different colored branches represent different data collection times. An enlargement of the more distributed branch of the *HA* sequence of the inactivated vaccine is shown on the left. (**B**) Line chart of the approved inactivated vaccines over time and a bar chart of the number of H9N2 influenza virus infections.

**Figure 3 viruses-14-01043-f003:**
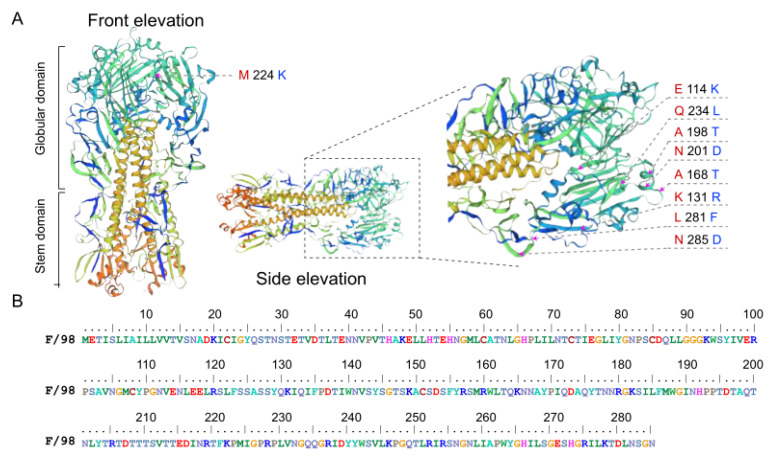
Homology modeling and amino acid sequence display of the HA protein of the H9N2 subtype influenza virus. (**A**) Diagram of the structural pattern of influenza virus. Three-dimensional structural prediction of the HA protein shows two distinct domains: globular domain and stem domain. The arrows represent the locations of mutation hotspots in the three-dimensional structure of the HA protein; the red font represents the amino acids in our sequence, and the blue font represents the amino acids mutated in the reference [20]. (**B**) The first 285 amino acid sequences of the HA protein of the F/98 strain.

**Figure 4 viruses-14-01043-f004:**
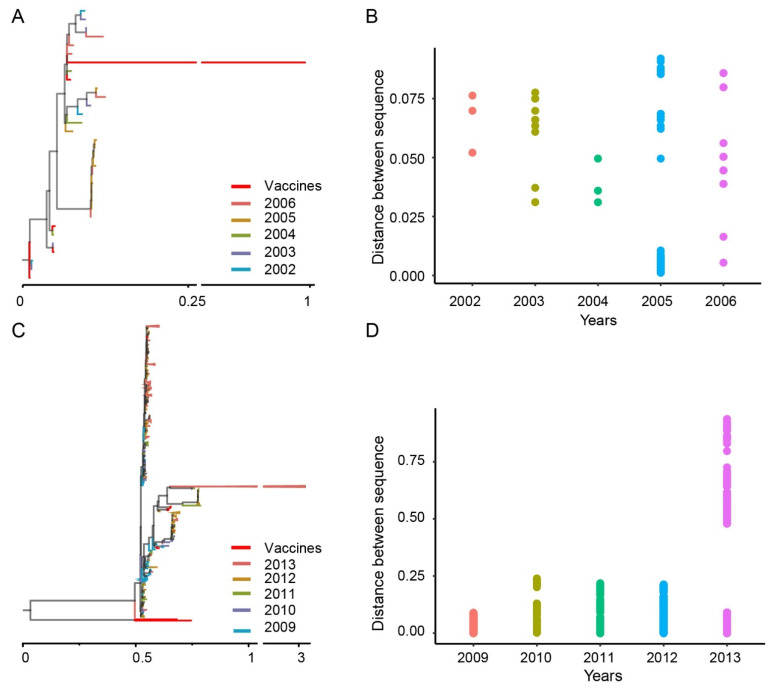
Genetic distance of the *HA* sequence of the H9N2 subtype avian influenza virus (AIV). (**A**) Phylogenetic tree of the *HA* gene of H9N2 AIVs. The color of the branch represents the collection date of the sequence, and the vaccine sequence is shown in red. (**B**) Distribution of genetic distances between *HA* sequences collected annually (excluding vaccine sequences). Genetic distances for different years are indicated in different colors. (**C**) Phylogenetic tree of the *HA* gene of the H9N2 AIV. The color of the branch represents the collection date of the sequence, and the vaccine sequence is shown in red. (**D**) Distribution of genetic distances between *HA* sequences collected annually (excluding vaccine sequences).

**Figure 5 viruses-14-01043-f005:**
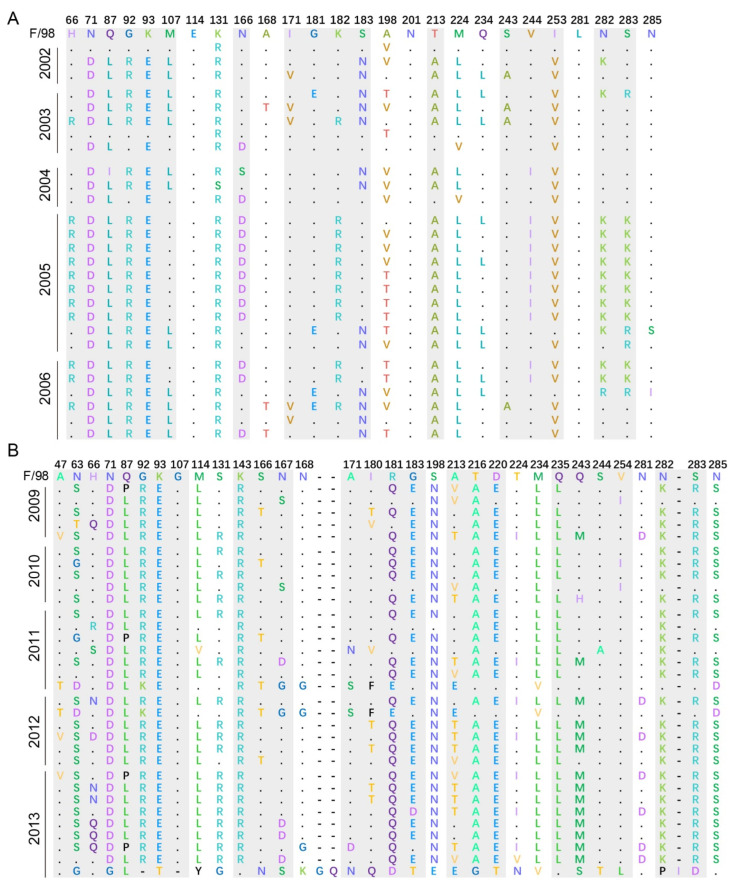
Distribution of mutational sites of the H9N2 subtype AIV at different time periods. (**A**) Sequence alignment of HA protein and F/98 protein of H9N2 subtype AIVs from 2002 to 2006. (**B**) Sequence alignment of HA protein and F/98 protein of H9N2 subtype AIVs from 2009 to 2013. The numbers in the first row indicate the location of the protein sites; the mutation sites reported are marked with a white background; the positive sites found in this study are shown with a light gray shade; and the strain name is on the left. The horizontal lines represent insertional mutations.

## Data Availability

All relevant data are included in this published article. The data sets used and/or analyzed during the current study are available from the corresponding author on request. The data that support the findings of this study have been deposited into the CNGB Sequence Archive (CNSA) of the China National GeneBank DataBase (CNGBdb) with accession number CNP 0002726 [32].

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
