# Peer review of "Hemagglutinin Gene Variation Rate of H9N2 Avian Influenza Virus by Vaccine Intervention in China"

_viruses, 2022, doi:10.3390/v14051043_

Round 1

Reviewer 1 Report

In this paper, Cao and colleagues have studied the gene variation rate of the HA of IAV and its correlation with vaccine use in China. The scientific question asked by the authors is interesting, the experiment were made in an intelligent fashion however some minor revisions and additional information are needed to improve the manuscript before publication.

Mat & Meth:

  • The authors should precise in the mat &meth which alignments were performed with nucleotide or amino acid sequences?

Results:

3.1: the authors should clearly define what is “birds and environment” (human is pretty clear!) and maybe replace it by “poultry and wild birds “

Improve the legend of figure 1: the map is not showing the distribution of the countries but the number of H9N2 cases in the different countries.

3.2: I would like to believe you but it is very difficult to visualize anything on the phylogenetic tree (right panel). The authors claimed that the genetic distance is increasing in 2013, 2017 and 2019… I can’t see it. In the enlarged part (left panel), the author should add color to enlight the distance between vaccine sequence and viral sequences.

What about the rate of evolution of HA sequences before H9N2 vaccination? Before 2002?

Is figure 2.B really useful? The authors didn’t discuss the link between the number of approved vaccines and the genetic distance increasing in 2013, 2017 and 2019.

The reviewer thinks that the 3.3 section should be replaced by 3.4. First sequences and then protein part displaying the location of the mutation hot spots for drifting.

Line 192: “mainly” distributed in the first 285 amino acids. Are mutations present in HA2 also?

Figure 3 is nice but weakly informative. Arrows are very difficult to distinguish. The hotspots are not represented on the sequence displayed in figure 3.B. Mutation hotspots are associated to only one amino acid (like GLU 114 for instance). Is there different mutants for each hotspot or only one? We could expect a variety of mutants for a given hotspot.

3.4: line 214: 32 HA sequences?

Same remark for the phylogenetic trees… unreadable

Figure 4.B and 4.C: each dot represents how many sequences? The difference of number of sequences engaged in the analysis could introduce a bias. I don’t know if I properly read this figure, but it seems that the distance between sequences is more important in A than in B. The three dots are already between 0.5 and 0.75 in 2002. The authors should explain more in details all this part and how they calculate the distance between sequences, and which sequences (between the sequences of the same year, a different year…)

3.5: The authors said that they “translated” the sequence in amino acids to perform alignment. Was it not the case for the previous analysis?  Do the authors consider silent mutations as evolution of the sequence?

Reviewer 2 Report

This manuscript addresses the variation of H9N2 AIV and this evolution by the vaccine intervention. This is the important topic for the control strategies against H9N2 AIV and it would affect the use of vaccines in the future.

 Therefore, results should include a reproducible, objective reporting of your finding.

■line 37

H9N2 virus has first detected in US in 1966, but not all viruses spread worldwide after then are the progeny. The virus surveillance would be too poor to understand the population of H9N2 virus worldwide then.

■line 44

This reference refers to H5N1 not to H9N2.

■line 86-94

The sequences of vaccines used in this study should be disclose.

■95-100

How many sequences did you use in the phylogenetic analysis?

How many bases of HA did you compare in the alignment?

In M&M all H9N2 sequences in China in 1979 to 2021 were downloaded and deduplicated according to the strain name. In Result 3.2., only 1,274 HA gene were used in phylogenetic analysis.

In contrast, I prepared the 6,321 sequences by same way except adding vaccine sequence.

The author should clarify the selection way of sequence used in the analysis.

■line 129

Which the sequence of vaccines in 28 one did you obtain? You should show 22 vaccines used in analysis.

■line 139

It’s not as if H9N2 virus in US in 1966 has spread to worldwide.

■line 174

It is difficult to speculate this conclusion if the amount used and coverage of vaccine were unknown. The author should provide this information.

■line 182-205

This section is not suitable in "Result" because the data is Su's result.

■line 253

Where is the “insertion mutation”?

Reviewer 3 Report

The science is good, the material and methods are well described, results and figures are clearly described, the discussion is well organized. Overall the report is well written and the conclusion is advanced with caution and leaves room for future advances which will make it possible to further investigate this track suggesting that vaccination (inactivating the virus) would have an impact on the genetic drift of the virus.

Minor corrections

L20: Change explore  to investigate.

L27-29. Clarify: the high frequency variation of vaccine strains contributed more to the rate of virus evolution than the low frequency variation of vaccine strains. Not clear what is high and what is low frequency variation.

L89. Clarify: All downloaded sequences were deduplicated

L130. Remove “obtained »

L266 : Change « were obtained from” to “represent”

L274: clarify “can be produced inside the body”. Do you mean “produce by the host”?

Round 2

Reviewer 2 Report

■Line 88

The author explained that “A total of 1436 sequences, including 1253 avian, 147 environmental and 36 human samples, were downloaded from the Global Initiative on Sharing Avian Influenza Data and the Influenza Virus Resource Database.”, but I can’t have the same results.

 The sequences in China from 1979 to 2021 which obtained from GISAID are as follows,

Avian: 8656 sequences

Environmental: 212 sequences

Human: 47 sequences

Though the download dates are different, the difference number of downloaded sequences can’t be overlooked. Additionally, the download date should be specified.

The phylogenetic analysis can lead different results depending on used sequences. Therefore, the sequences used in phylogenetic analysis should be objectively selected. The author must explain how to select the sequence.

In first comment, I thought that the filtration way of downloaded sequences was different. But, the author's answer show that the download numbers are different.

■Line 192

It is difficult to speculate this conclusion if the amount used and coverage of vaccine were unknown. The author should provide or discuss this information. The answer of the author to this comment was misplaced because the necessary information to this comment is not the source of the sequence.

The author's conclusion in this section is "there is some correlation between the evolutionary distance of H9N2 subtype AIV genome and the continuous use of inactivated vaccines." But, the evolution in 2013 seems to be resulted that the number of cases were many. It is contrived that the author didn't mention to the evolution after 2014.

■Table S1 and Figure 2A

Why "SH" and "H2" are in Figure 2?

The author showed that these were not used in analysis in Table S1, didn't it?
